# Beta variant COVID-19 protein booster vaccine elicits durable cross-neutralization against SARS-CoV-2 variants in non-human primates

Vincent Pavot [1,6], Catherine Berry [1,6], Michael Kishko [2], Natalie G. Anosova[2], Lu Li[2], Tim Tibbitts[2], Dean Huang[2], Alice Raillard[1], Sylviane Gautheron [1], Cindy Gutzeit[3], Marguerite Koutsoukos [4], Roman M. Chicz[5] & Valerie Lecouturier [1] ✉

The rapid spread of the SARS-CoV-2 Omicron subvariants, despite the implementation of booster vaccination, has raised questions about the durability of protection conferred by current vaccines. Vaccine boosters that can induce broader and more durable immune responses against SARS-CoV-2 are urgently needed. We recently reported that our Beta-containing protein-based SARS-CoV-2 spike booster vaccine candidates with AS03 adjuvant (CoV2 preS dTM-AS03) elicited robust cross-neutralizing antibody responses at early time-points against SARS-CoV-2 variants of concern in macaques primed with mRNA or protein-based subunit vaccine candidates. Here we demonstrate that the monovalent Beta vaccine with AS03 adjuvant induces durable cross-neutralizing antibody responses against the prototype strain D614G as well as variants Delta (B.1.617.2), Omicron (BA.1 and BA.4/5) and SARS-CoV-1, that are still detectable in all macaques 6 months post-booster. We also describe the induction of consistent and robust memory B cell responses, independent of the levels measured post-primary immunization. These data suggest that a booster dose with a monovalent Beta CoV2 preS dTM-AS03 vaccine can induce robust and durable cross-neutralizing responses against a broad spectrum of variants.

Over the past 2 years, the world has witnessed SARS-CoV-2 infection associated with the emergence of circulating variants of concern (VOCs) arising from mutations during virus replication that can be advantageous regarding viral fitness and/or immune escape. To combat the latest Omicron subvariants that display increasing immune escape, booster vaccines based on predominantly circulating variants have been developed[1]. This strategy would mimic what has been done successfully with vaccines based on the ancestral SARS-CoV-2 strain (Wuhan). However, it relies on many assumptions: (1) rapid deployment of updated variant-specific vaccines before a new escaping variant displaces the current circulating strain; (2) induction of a potent variant-specific immune response in already primed populations; and (3) potentially several booster vaccinations per year to follow the epidemiology. While the last assumption is unlikely, the first two are now challenged. In terms of the vaccine updating process, circulating variants have been replaced every 3–4 months globally, since BA.1

[1]Sanofi, Marcy l'Etoile, France. [2]Sanofi, Cambridge, MA, USA. [3]GSK, Rixensart, Belgium. [4]GSK, Wavre, Belgium. [5]Sanofi, Waltham, MA, USA. [6]These authors contributed equally: Vincent Pavot, Catherine Berry. ✉e-mail: valerie.lecouturier@sanofi.com

mRNA-primed cohort

Protein subunit-primed cohort

**Fig. 1 | Study schema.** In the mRNA-primed cohort (left), three groups of four cynomolgus macaques were immunized intramuscularly with mRNA COVID-19 (ancestral D614) vaccine candidates on day 0 (D0) and on day 21 (D21). In subunit-primed cohort (right), three groups of five rhesus macaques were immunized intramuscularly with CoV2 preS dTM-AS03 (ancestral D614) vaccine candidates on D0 and D21. Both cohorts were boosted 7 months post-dose 1 with monovalent (ancestral or Beta) or bivalent (ancestral + Beta) CoV2 preS dTM-AS03. Humoral immune responses were assessed up to 6 months post-booster.

emergence at the end of November 2021. Although the duration of mRNA vaccine development has been considerably reduced, they may not cope with the current epidemiology. With respect to the second assumption on immunogenicity, evidence indicates that immune imprinting due to prior immunization or infection restricts the repertoire of neutralizing antibody (NAb) responses, mainly to the cross-reactive one[2]. An alternative approach could be the selection of a highly immunogenic antigen capable of modulating the immunodominant responses of the parental vaccine while preserving the diversity of the neutralizing responses to target multiple VOCs, including the currently dominant variant. One antigen candidate fulfilling these conditions is the Beta variant spike[3].

In parallel to the immune escape, evidence suggests that the protective efficacy of mRNA-based booster vaccination against Omicron infection and severity is waning rapidly, prompting to evaluate alternative platforms for durability[4].

We previously reported vaccine efficacy with the CoV2 preS dTM-AS03 (ancestral D614) candidate in non-clinical models[5] and demonstrated safety and immunogenicity in humans in a Phase-2 clinical trial[6]. After the emerging epidemic caused by the Beta variant (B.1.351) in South Africa from September 2020 to May 2021, we tested recombinant protein vaccine candidates with AS03-adjuvant (monovalent and bivalent) based on the Beta variant spike in preclinical models and clinical trials. In nonhuman primates (NHPs) previously primed with the ancestral D614 spike antigen (mRNA or recombinant protein), our Beta-based recombinant protein boosters have shown broad cross-reactivity across SARS-CoV-2 VOCs (Alpha, Beta, Gamma, Delta, and Omicron BA.1) and SARS-CoV-1 at early time points post-booster[7].

These observations have been confirmed in two Phase 3 clinical trials (NCT04762680 and NCT05124171), demonstrating that our Beta-containing booster vaccine candidate delivered a strong immune response against VOCs, including Omicron, in adults primed with mRNA vaccines and conferred strong efficacy against symptomatic infection in adults and participants previously infected with SARS-CoV-2[3,8–10].

In the present work, we report the durability of the cross-neutralizing antibody responses in NHPs at 6 months post-booster and the effect on memory B-cells after a third immunization with various formulations, namely ancestral (D614), variant (Beta), and bivalent (ancestral + Beta) CoV2 preS dTM-AS03 in mRNA- and subunit-primed macaques. We show that the increase of cross-neutralizing antibody responses is prolonged up to 6 months, with NAb titers against Omicron BA.1, Omicron BA.4/5, and SARS-CoV-1 detected in all macaques immunized with the monovalent Beta-AS03 formulation. Importantly, the spike-specific memory B cells measured at 3 months were augmented after the booster immunization, especially in macaques with low memory B cell responses after the primary immunization.

The potent and prolonged booster effect up to 6 months against VOCs provided by a booster dose with the Beta-containing

recombinant CoV2 preS dTM-AS03 vaccine candidate in NHPs, combined with the encouraging results from clinical trials presents unique benefits for future booster vaccination campaigns[3,8–10].

## Results

### Beta-containing AS03-adjuvant vaccine booster delivers durable broad neutralizing antibody responses against variants of concern

Using a lentivirus-based pseudovirus-neutralization assay, we assessed NAb responses at 6 months post-booster (Fig. 1), in mRNA and subunit-primed NHP cohorts for which we already reported the results up to 3 months post-booster[7].

In mRNA-primed macaques, the geometric mean titers (GMTs) at 6 months post-booster (D176) ranged from 400 to 1951 against prototype D614G, 81 to 575 against Beta, 165 to 514 against Delta, 19 to 226 against Omicron BA.1, and 59 to 132 against SARS-CoV-1 (Fig. 2). At 6 months, 100% of the macaques immunized with the Beta-containing boosters still had detectable NAb titers (>LOQ) against all VOCs and SARS-CoV-1, except for one macaque against Omicron BA.1 and SARS-CoV-1 in the bivalent vaccine group. This could be explained by the lower amount of Beta antigen in the bivalent D614 + Beta-AS03 vaccine (2.5 μg) compared to the monovalent Beta-AS03 vaccine (5 μg).

In subunit-primed macaques, the GMTs at 6 months post-booster (D178) ranged from 1082 to 2600 against prototype D614G, 243 to 617 against Beta, 419 to 870 against Delta, 73 to 263 against Omicron BA.1, and 104 to 250 against SARS-CoV-1 (Fig. 2). In the groups immunized with the monovalent ancestral D614-AS03 and Beta-AS03 vaccines, 100% of the macaques had detectable NAb titers against all VOCs and SARS-CoV-1 at 6 months.

Decay of NAb titers against the D614G, Beta, and Delta variants were modeled for each strain using a regression linear model or quadratic model with plateau, when applicable. In macaques primed with the mRNA vaccine, a slight decrease of NAb titers with time was observed, regardless of variants, similar for all vaccine formulations and estimated between 2.5 and 5-fold/100 days.

In macaques primed with the subunit vaccine, a slight decrease of D614G NAb titers with time was observed in the monovalent Beta-AS03 group estimated at 4-fold/100 days. In the two other groups (monovalent ancestral and bivalent), the D614G NAb titers stabilized around 6 months post-booster at GMTs ranging from 1082 to 2600. In vaccine groups with adjuvant, the Beta and Delta NAb titers stabilized around 6 months post-booster.

Omicron BA.1 and SARS-CoV-1 NAb titer decay was not modeled as no data were available at intermediate time points.

Neutralizing titers against Omicron BA.4/5 were only analyzed at 6 months post-booster (Fig. 3). In both cohorts, data show that 100% of the macaques immunized with the monovalent Beta-AS03 formulation still have detectable neutralizing titers with a GMT of 100. The D614-AS03 formulation induced 100% responders only when used as a

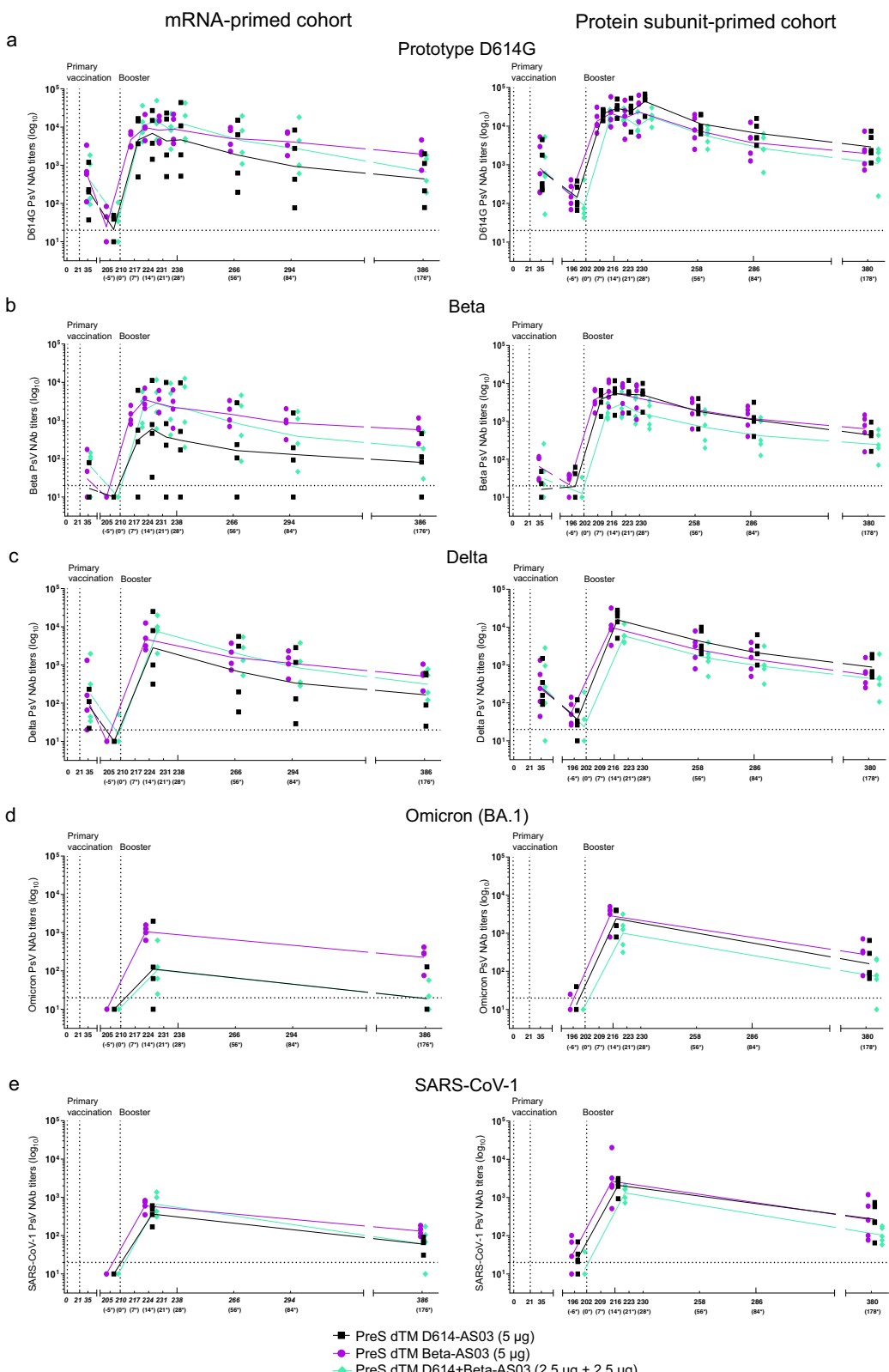

**Fig. 2 | Kinetics of booster-neutralizing antibody responses in primed macaques.** Pseudovirus-neutralizing antibody against **a** the prototype D614G, variants of concern **b** Beta, **c** Delta, and **d** Omicron (BA.1), and **e** SARS-CoV-1 were assessed up to D176 (mRNA-primed macaques, $n = 4$) or day 178 (protein-primed macaques, $n = 5$) post-booster with ancestral D614 vaccine, Beta vaccine, or bivalent ancestral + Beta vaccine. Individual macaque data are shown. Connecting lines represent geometric mean titers (GMTs). Horizontal dotted lines represent the limit of quantification of the assay. *Timepoints relative to booster injection day.

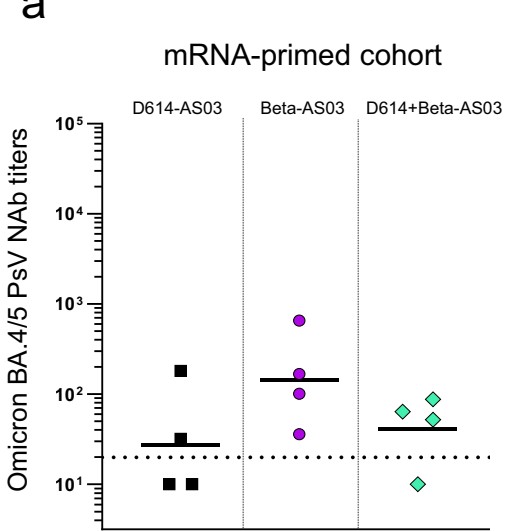

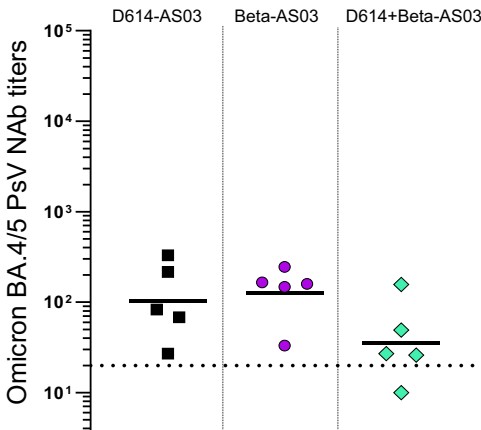

**Fig. 3 | Neutralizing antibody responses against SARS-CoV-2 Omicron BA.4/5 at 6 months post-booster in primed macaques.** Pseudovirus-neutralizing antibodies against Omicron BA.4/5 variant were assessed at **a** day 176 (mRNA-primed macaques, $n = 4$) or **b** day 178 (protein-primed macaques, $n = 5$) post-booster with ancestral D614 vaccine, Beta vaccine, or bivalent ancestral + Beta vaccine. Individual macaque data were shown with geometric mean titers (GMTs). Horizontal dotted lines represent the limit of quantification of the assay.

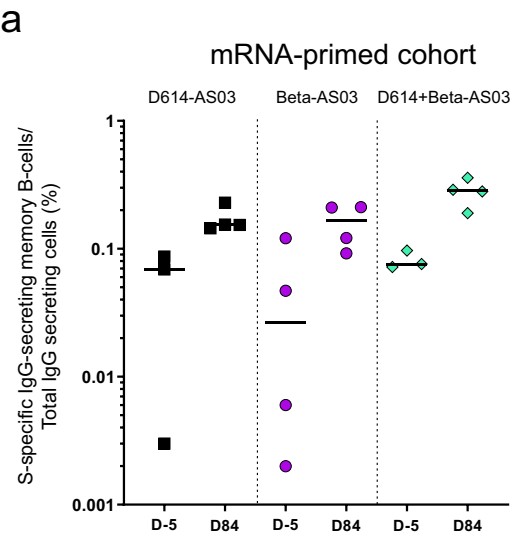

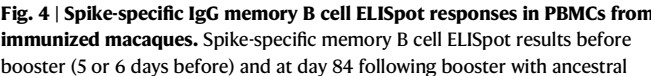

**Fig. 4 | Spike-specific IgG memory B cell ELISpot responses in PBMCs from immunized macaques.** Spike-specific memory B cell ELISpot results before booster (5 or 6 days before) and at day 84 following booster with ancestral D614 vaccine, Beta vaccine or bivalent ancestral + Beta vaccine in macaques previously immunized with **a** mRNA vaccine candidate ($n = 4$) or **b** protein-subunit vaccine candidate ($n = 5$). Bars = Medians.

booster in the protein subunit-primed cohort. The bivalent D614 + Beta-AS03 showed lower immunogenicity against Omicron BA.4/5 in both cohorts, probably due to a lower antigen dose (2.5 μg).

Spike-specific IgG ELISA titers post-booster are shown in Supplementary Fig. 1.

### Boosting with Beta-containing vaccines induced high Spike-specific IgG memory B-cells

Because reactive memory B-cells—that do not constitutively secrete antibodies—might be more durable than circulating antibodies and thus crucial for long-term immunity, we assessed circulating memory B cells in the blood. Spike-specific memory B-cells were quantified by ELISpot assay in peripheral blood mononuclear cells (PBMCs)

collected from the immunized macaques at baseline (before booster) and on Day 84 after booster to assess the vaccine-induced long-term reactive humoral memory responses. Spike-specific IgG memory B-cells were detected at baseline (6 months post-primary immunizations), in all groups, at medians ranging from: 0.02 to 0.2% of total IgG-secreting cells (Fig. 4). Spike-specific memory B-cells increased after the booster vaccine, especially in the low-responding macaques leading to homogeneous and robust responses in all macaques (medians ranging from: 0.15 to 0.2%). All formulations performed similarly.

## Discussion

The protection conferred by currently available SARS-CoV-2 vaccines based on the ancestral strain (Wuhan) declines substantially within a

few months of vaccination, particularly in the context of the Omicron variants, prompting health authorities to solicit more effective, modified booster vaccines.

Based on clinical data with prior bivalent mRNA vaccines and preclinical data on BA.5 bivalent vaccines, some regulatory authorities have recommended bivalent vaccines incorporating the latest Omicron descendent lineage and ancestral virus for adapted versions of vaccines already in use, whereas other regulatory authorities are considering a broader approach[11].

At the time of writing, clinical data supporting the bivalent BA.5 containing mRNA vaccines are still awaited, but some preliminary data on a fourth dose in humans suggest no significant benefits over the parental vaccines on the NAb titers against all variants tested, while a significant decrease was observed on very distant sarbecoviruses[12]. Whether these results are confirmed might impact the future decision on COVID-19 booster recommendation.

We previously showed that, in macaques primed with Sanofi's mRNA or protein-based subunit vaccine candidates (ancestral D614), one booster dose of Beta-containing CoV2 preS dTM-AS03 (monovalent or bivalent ancestral + Beta), significantly boosted the pre-existing NAbs against the D614G strain from 177- to 370-fold and elicited high cross-neutralizing antibodies covering five SARS-CoV-2 VOCs (Alpha, Beta, Gamma, Delta, and Omicron BA.1) and SARS-CoV-1[7].

In the present study, we show that those NAb responses were maintained up to 6 months post-booster with ancestral D614 or Beta-containing vaccine formulations. Importantly, NAb titers were measured in all macaques immunized with the Beta CoV2 preS dTM-AS03 in both cohorts (mRNA- and subunit-primed) at 6 months, while some animals immunized with the bivalent D614 + Beta vaccine had undetectable titers against Omicron BA.1, BA.4/5, and SARS-CoV-1 at 6 months.

In macaques primed with the mRNA vaccine, we observed a slight decrease of NAb titers with time after the booster, similar for all vaccine formulations, with the same kinetics regardless of variants, estimated between 2.5 and 5-fold/100 days. In macaques primed with the subunit vaccine, we observed a slight decrease of D614G NAb titers with time in the monovalent Beta groups estimated at 4-fold/100 days. In the other groups, the D614G NAb titers stabilized around 6 months post-booster. In all groups, the Beta and Delta NAb titers stabilized around 6 months post-booster.

Furthermore, in our studies, the extended breadth of neutralization to SARS-CoV-1, Omicron BA.1, and Omicron BA.4/5 was confirmed until 6 months post-booster and are consistent with previous observations made with the same AS03 adjuvant with an alternative protein[13,14]. The neutralizing titers induced against SARS-CoV-1 were similar to the titers against Omicron, although the SARS-CoV-1 spike bears only 72% homology with the SARS-CoV-2 spike.

Meta-analyses have established high correlations between the standardized mean titer and vaccine efficacy, and the neutralizing antibody titer has consistently been shown to be a mechanistic correlate of protection (CoP) in humans[15,16] and challenge studies in nonhuman primates[17,18]. The evidence strongly supports the designation of the neutralizing antibody titer as a CoP but no thresholds have been identified yet, possibly because COVID-19 is caused by a mucosal infection and serological antibody titers are not predictive enough of mucosal titers[15] or because immune evasion of variants would require to match the assay to the infecting variant for each case.

Interestingly, in the study from ref. [17] in rhesus macaques, it was shown that relatively low neutralizing antibody titers (threshold: pseudovirus NAb titers of 43) are sufficient for protection against SARS-CoV-2. Considering this threshold, we can be confident that the monovalent Beta vaccine formulation would induce protection against all tested variants up to 6 months post-booster.

Clinical studies have demonstrated that the NAb titers observed at peak immunity after mRNA primary vaccination wane rapidly and substantially, eliciting a constant decrease up to 8 months[19,20]. A significant antibody decay was also observed after mRNA third or fourth dose booster vaccination, associated with a gradual waning of vaccine efficacy against Omicron infection[21]. Neutralizing antibody titers after adenovirus type 26 (Ad26) vector vaccination are more durable over time, but peak at lower levels[20]. The serum concentration of NAbs needed to provide durable protection from infection and transmission remains unclear but is likely higher for the more infectious SARS-CoV-2 variants compared to the original Wuhan strain. The high peak titers combined with a slow Ab decay observed with our Beta CoV2 preS dTM-AS03 vaccine candidate is expected to result in higher titers at later time points than after booster immunization with currently approved vaccines and might thus provide a clinical benefit. The stability of the NAb titers observed between 3 and 6 months after the booster immunization illustrates improved durability of the circulating antibodies compared to the current mRNA vaccines used in primary vaccination or as boosters[22–24].

Moreover, all vaccine formulation boosters expanded the Spike-specific IgG-secreting memory B cell populations that were generated post-primary vaccination. Spike-specific memory B-cells were increased mainly in the low-responding macaques, resulting in homogeneous and robust responses post-booster. These pathogen-experienced memory B-cells induced by vaccination do not constitutively secrete antibodies but may have a critical role in controlling the viral replication when activated, such as in cases of breakthrough infections[25,26]. For COVID-19 vaccines, although serum antibody titers may wane rapidly, memory B-cells are highly durable and may contribute to protection from disease along with memory T cells[23,27].

These data provide information about alternative boosting strategies to address the risks associated with SARS-CoV-2 VOCs and waning immunity. Based on these encouraging results, Beta-containing vaccine formulations are being evaluated as booster vaccines in a clinical trial, and preliminary results confirm and extend our observations in primed NHPs by showing a superiority of the Beta-containing vaccine when compared to the ancestral D614 vaccine for the neutralization of a wide range of variants (NCT04762680, NCT04904549, and NCT05124171)[8–10].

In conclusion, our Beta-containing AS03-adjuvant vaccine booster demonstrated a potential to induce durable and high cross-neutralizing Ab responses against a broad spectrum of SARS-CoV-2 variants and SARS-CoV-1 for at least 6 months with limited Ab decay as well as robust memory B-cells expansion in NHPs, independent of the vaccine platform used for primary vaccination.

## Methods

All animal experiments were carried out in compliance with all pertinent US National Institutes of Health regulations and were approved by the Institutional Animal Care and Use Committee (IACUC) from the University of Louisiana at Lafayette New Iberia Research Center (IACUC number 2020-8733-013).

### Vaccines

For primary immunization, the mRNA vaccines were SARS-CoV-2 pre-fusion spike constructs 2 P, GSAS, 2 P/GSAS, 2 P/GSAS/ALAYT, and 6 P/GSAS described in ref. [28], the subunit vaccines were CoV2 preS dTM-AS03 vaccines, where the antigens were produced using the phase-I/-II manufacturing process, 1.3- and 2.6-μg doses, or using an intermediate manufacturing process, 2.4 μg dose.

For the booster, the CoV2 preS dTM derived from the ancestral strain (D614) and the Beta variant were produced using an optimized purification process to ensure a minimum of 90% purity.

**Table 1 | Description of the reporter-virus particles (RVPs) used in the pseudovirus-neutralization assay**

| Pango Lineage | WHO Name | Sequence source | Catalog number | Lot | Mutations relative to Wuhan ancestral |
|---|---|---|---|---|---|
| B.1 | N/A | – | RVP-702G | CG-129A | D614G |
| B.1.351 | Beta | Tegally et al. 2020[29] | RVP-724G | CG-180A | L18F, D80A, D215G, ΔL242/A243/L244, R246I, K417N, E484K, N501Y, D614G, A701V |
| B.1.617.2 | Delta | cov-lineages.org | Custom | CG-233A | T19R, G142D, E156G, ΔF157/R158, L452R, T478K, D614G, P681R, D950N |
| B.1.1.529 | Omicron (BA.1) | EPI_ISL_6841980 | RVP-768G | CG-296A | A67V, Δ69-70, T95I, G142D/Δ143-145, Δ211/L212I, ins214EPE, G339D, S371L, S373P, S375F, K417N, N440K, G446S, S477N, T478K, E484A, Q493R, G496S, Q498R, N501Y, Y505H, T547K, D614G, H655Y, N679K, P681H, N764K, D796Y, N856K, Q954H, N969K, L981F |
| B.1.1.529 | Omicron (BA.4/5) | UTO31503.1 | RVP-774G | CG-352A | T19I, LPPA24S, HV69del, G142D, V213G, G339D, S371F, S373P, S375F, T376A, D405N, R408S, K417N, N440K, L452R, S477N, T478K, E484A, F486V, Q498R, N501Y, Y505H, D614G, H655Y, N679K, P681H, N764K, D796Y, Q954H, N969K |
| NA | SARS-CoV-1 | P59594.1 | RVP-801G | SG-115B | 28% differences |

*N/A* not applicable.

The antigens were formulated in monovalent or bivalent formulations with AS03 adjuvant. The CoV2 preS dTM was produced from a Sanofi proprietary cell culture technology based on the insect cell−baculovirus system, referred to as the Baculovirus Expression Vector System (BEVS). The CoV2 preS dTM (ancestral D614) sequence was designed based on the Wuhan YP_009724390.1 strain S sequence, modified with 2 prolines in the S2 region, deletion of the transmembrane region, and addition of the T4 foldon trimerization domain. The CoV2 preS dTM (Beta) was designed based on the Beta (B.1.351) sequence (GISAID Accession EPI_ISL_1048524) and contains the same modifications.

AS03 is a proprietary adjuvant system composed of α-tocopherol, squalene, and polysorbate-80 in an oil-in-water emulsion manufactured by GSK. Vaccine doses were formulated by diluting the appropriate dose of preS dTM with PBS−tween to 250 μL, then mixing with 250 μL of AS03, followed by inversion five times for a final volume of 500 μL. Each dose of AS03 contains 11.86 mg of α-tocopherol, 10.69 mg of squalene, and 4.86 mg of polysorbate-80 (Tween 80) in PBS.

### Animals and study design
Animal experiments were carried out in compliance with all pertinent US National Institutes of Health regulations and were conducted with approved animal protocols from the Institutional Animal Care and Use Committee (IACUC) at the research facilities. NHP studies were conducted at the University of Louisiana at Lafayette New Iberia Research Center.

Two cohorts of vaccinated NHPs received a booster immunization after randomizing each group within a cohort based on their baseline characteristics (Fig. 1).

In the mRNA-primed cohort, six adult male and six adult female Mauritius cynomolgus macaques (*Macaca fascicularis*) aged 4–10 years, selected based on their responses to the primary vaccination, were randomly allocated to three groups of four animals according to their baseline characteristics.

In the subunit-primed cohort, 15 adult male Indian rhesus macaques (Macaca mulatta) aged 4−7 years were randomly allocated to three groups of five animals. In the priming phase, animals received two immunizations of either Sanofi's mRNA COVID (ancestral D614) experimental candidate vaccines or CoV2 preS dTM-AS03 (ancestral D614) vaccine through the intramuscular route in the deltoid at day 0 and day 21. Seven months after the primary immunization, both cohorts were immunized with CoV2 preS dTM (ancestral)−AS03, CoV2 preS dTM (Beta)−AS03, and a bivalent CoV2 preS dTM (ancestral + Beta)−AS03. All groups received a total dose of 5 μg of CoV2 preS dTM antigen. All immunologic analyses were performed blinded on serum collected at 7, 14, 21, 28, 56, 84 days, and 6 months post-boost injection for D614G and Beta

seroneutralizations; on D14, 56, 84, and 6 months for Delta; on D14 and 6 months for Omicron (BA.1), Omicron BA.4/5, and SARS-CoV1. Animal studies were conducted in compliance with all relevant local, state, and federal regulations, and were approved by the New Iberia Research Center.

### Pseudovirus-based virus neutralization assays
Serum samples were diluted 1:4 or 1:20 in media (FluoroBrite™ phenol-red-free DMEM + 10% FBS + 10 mM HEPES + 1% PS + 1% GlutaMAX™) and heat-inactivated at 56 °C for 30 min. Further, a twofold, 11-point, dilution series of the heat-inactivated serum were performed in media. Diluted serum samples were mixed with reporter-virus particle (RVP)-GFP (Integral Molecular) listed in Table 1 diluted to contain ~300 infectious particles per well and incubated for 1 h at 37 °C. Ninety-six well plates of ~50% confluent 293T-hsACE2 clonal cells (Integral Molecular, Cat# C-HA102) in 75 μL volume were inoculated with 50 μL of the serum + virus mixtures and incubated at 37 °C for 72 h. At the end of the 72-h incubation, plates were scanned on a high-content imager and individual GFP-expressing cells were counted. The neutralizing antibody titer was reported as the reciprocal of the dilution that reduced the number of virus plaques in the test by 50%.

### Enzyme-linked immunosorbent assay (ELISA)
Nunc microwell plates were coated with SARS-CoV S-GCN4 protein (GeneArt, expressed in Expi 293 cell line) at 0.5 μg/mL in PBS at 4 °C overnight. Plates were washed three times with PBS−Tween 0.1% before blocking with 1% BSA in PBS−Tween 0.1% for 1 h. Samples were heat-inactivated at 56 °C for 30 min and plated at a 1:450 initial dilution followed by threefold, seven-point serial dilutions in blocking buffer. Plates were washed three times after 1-h incubation at room temperature before adding 50 μL of 1:8000 Goat anti-human IgG (Jackson Immuno Research, CAT# 109-036-098) to each well. Plates were incubated at room temperature for 1 h and washed thrice. Plates were developed using Pierce 1-Step Ultra TMB-ELISA Substrate Solution for 6 min and stopped by TMB STOP solution. Plates were read at 450 nm in a SpectraMax® plate reader, and the data analyzed using Softmax® Pro 6.5.1 GxP software and the proprietary software, Sanofi Universal Exporter 2.1. Antibody titers were reported as the highest dilution that is equal to 0.2-OD cutoff.

### Enzyme-linked immunospot (ELISpot)
Memory B cells were analyzed using Human IgG Single-color B cell ELISpot kit (CTL, CAT# NC1911372). Cryo-preserved PBMCs were quickly thawed in a 37 °C water bath. A Fetal Calf Serum (FCS) /DNAse I (200 unit/mL) mixture was slowly added to PBMCs, before being transferred to a complete cell culture medium (CM) (RPMI 1640 with

10% FCS and antibiotic cocktail). After centrifugation and resuspension into 6 mL of CM, PBMCs were transferred into 6-well plates and incubated at 37 °C with 5% of $CO_2$ for 1 h. Then, B-Poly-S™ was added at 1:1000 dilution for cell stimulation, for up to 4 days at 37 °C with 5% of $CO_2$.

Pre-stimulated PBMCs were harvested and centrifuged at $433 \times g$ for 5 min at room temperature (RT). After washing, PBMCs were counted using Guava® easyCyte cell counter and the cells were adjusted to the desired concentration with CM.

Ninety-six well-plates with PVDF membrane were permeabilized with 15 µl of 70% ethanol for a maximum of 1 min, then washed three times with sterile phosphate-buffered saline (PBS) 1X before being coated with 80 µL of human Ig capture antibody (Ab) diluted at 1:50 or with SARS-CoV2 S-GCN4 protein, Wuhan, (GeneArt, expressed in Expi 293 cell line) at 4 µg/mL, or with PBS. The plates were incubated overnight at 4 °C and then washed three times with sterile PBS and blocked with CM for 1 h at RT. CM was then removed and PBMCs were added at $3 \times 10^5$ cells in the S-GCN4 protein and PBS-coated wells, and at $5 \times 10^3$ cells in the Ig capture Ab-coated wells, under 100 µL/well. Each condition was tested in duplicate, and plates were incubated for 18 h at 37 °C with 5% $CO_2$.

To reveal antibody-secreting cells, plates were first washed twice with PBS, then twice with 0.05% Tween PBS, and then 80 µL of anti-human IgG detection solution was added. After 2 h of incubation at RT, plates were washed three times with 0.05% Tween PBS, and 80 µL of a tertiary solution was added. Plates were incubated for 1 h at RT, then washed twice with 0.05% Tween PBS and twice with distilled water. Eighty microliters of a blue developer solution was added and incubated at RT for 15–20 min. The reaction was stopped by rinsing the plate membrane with water and decanting it three times. The plates were air-dried, then scanned and read using Cytation 7 analyzer. The number of spots with the PBS only (background) was subtracted from the number of S-specific or total IgG spots. The results are expressed as S-specific IgG-secreting memory B cells/million PBMCs and as % of S-specific IgG-secreting memory B cells among all circulating IgG-secreting memory B cells.

### Statistical analyses

For both mRNA-primed and CoV2 preS dTM-AS03-primed cohorts, at the time of the assignment, the characteristics at baseline (sex, age, and weight) were balanced to have comparable groups. The pseudovirus-neutralizing titers were also taken into account as well as the previous vaccine groups. ELISA titers and neutralizing titers were $log_{10}$-transformed prior to statistical analysis.

Due to the differences in species and gender repartition between both cohorts, no direct comparison was performed on immune responses between the two cohorts.

The time-effects post-booster (nAb decay) were modeled by-product for a given PsV variant using a linear regression model or quadratic model with plateau when applicable. If the decay was linear for all the products, an ANCOVA (Analysis of Covariance) model was used, the product was considered as a categorical factor and the time a continuous and repeated factor. When the interaction between group and time was statistically non-significant (<10%), meaning that all products have the same behavior according to the time, a common slope model was used. When the interaction was statistically significant, the slopes were calculated for each product separately. The models were considered acceptable if the studentized residual distribution was considered normal.

Analyzes were performed using SEG SAS v9.4®.

### Reporting summary

Further information on research design is available in the Nature Portfolio Reporting Summary linked to this article.

## Data availability

Accession codes and web links for publicly available datasets are provided in the manuscript. The source data generated in this study are provided as Supplementary Material.

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

## Acknowledgements

The authors thank Jon Smith for coordinating the production and providing the vaccine antigens for the study, Caroline Patriarca Ruat for project management, Tong-Ming Fu for coordinating the sharing of mRNA-primed macaques, and Carlos Diaz-Granados, Stephen Savarino, and Saranya Sridhar for critical discussions on the study designs and data analysis. The authors thank Julie Piolat for statistical analyses support. The authors also thank Hanson Geevarghese and Saili Dharadhar (Sanofi) for manuscript coordination and editorial assistance, respectively. This work was done in collaboration with GSK, who provided access to, and use of, the AS03 adjuvant system. This study was funded in whole or in part by Sanofi and by US federal funds from the Biomedical Advanced Research and Development Authority (BARDA), Administration for Strategic Preparedness and Response at the US Department of Health and Human Services under Contract # HHSO100201600005I, and in collaboration with the US Department of Defense Joint Program Executive Office for Chemical, Biological, Radiological and Nuclear Defense under Contract # W15QKN-16-9-1002.

## Author contributions

V.P., C.B., V.L., T.T., N.G.A., A.R., S.G., R.M.C., C.G., and M.Ko. contributed to the concept or design of the study. V.P., C.B., D.H., V.L., N.G.A., M.K., and L.L. collected and analyzed data. T.T. provided study coordination. C.G. and M.K. contributed critical reagents. D.H., N.G.A., C.B., V.P., V.L., M.K., T.T., A.R., S.G., R.M.C., C.G., and M.Ko. were involved in the analysis and interpretation of the data. V.L. and R.M.C. provided supervision. V.P. and C.B. drafted the first paper and all authors critically reviewed the paper versions. All authors had full access to the data and approved the paper before it was submitted by the corresponding authors.

## Competing interests

All authors have declared the following interests: C.B., V.P., M.K., L.L., N.G.A., T.T., D.H., A.R., S.G., V.L., and R.M.C. are Sanofi employees and may own Sanofi shares. M.Ko. and C.G. are employees of the GSK group of companies and report ownership of GSK shares.
