## [Peer Review File · Nature Communications]

REVIEWER COMMENTS

Reviewer #1 (Remarks to the Author):

Pavot et al reported the use of a preS stabilized recombinant spike protein of SARS-CoV-2 formulated in AS03 adjuvant (CoV2 preS dTM-AS03) as a booster vaccine against COVID-19. The work is the continuation of previous description of CoV2 preS dTM-AS03 vaccine and aims at demonstrate the broadness and durability of the induced neutralizing antibodies (nAb) in serum. For that purpose, three groups of non-human primates (NHP) primed with ancestral D614 spike expressing mRNA (two injections 4 weeks apart) were boosted six months later with either D614 CoV2 preS dTM-AS03, or Beta strain CoV2 preS dTM-AS03, or a mix of these two proteins. In addition, three other groups of NHP primed with two injections (4 weeks apart) of D614 CoV2 preS dTM-AS03, were boosted six months later as above with the same three approaches. The inclusion of Beta variant spike in the booster strategy represents a rational approach due to its immunogenicity and the potential capacity to elicit antibodies neutralizing an extended diversity of variants of concerns (VOC) which seems to be confirmed by the reported results. The authors demonstrate that 6 months post boost, most the NHP maintained significant levels of serum neutralizing activities against Alpha, Beta, Delta and BA.1 VOC. Remarkably, a majority of animals also developed nAb against more distant SARS-CoV-1. Circulating memory B cells in the blood that are specific of the spike protein have been assessed as a mechanism to explore the capacity of the vaccine to induce durable responses. Interestingly, spike specific B cells still datable six months post prime. Booster injection with CoV2 preS dTM-AS03 appeared to induce an increase of the specific IgG secreting cells in all groups at similar levels, including in animals with pre-boost low frequency of cells. This also confirms that protein boost can improve vaccine response whatever the priming strategy. The data presented here is certainly relevant to field and deserves to be considered for publication since the vaccine is already at advanced stages of the development and because the urgent need for new vaccines capable of inducing sustained cross-protective responses while avoiding multiple booster vaccine injections per year.

Specific comments:w

1. In mRNA-primed macaques, at M6 post boost, GMT ranged from 19 to 1951 depending on the tested variant. Not surprisingly lowest responses are observed against BA.1 VOC. What is the threshold of neutralizing antibodies the authors consider critical to maintain a protective response? This may be at least commented in the discussion section by comparing to the large amount of currently published data using similar classical pseudotyped lentivirus based assay for testing neutralization.

2. We may regret that animals have not been challenged by one of the recent VOC to assess protection and to try to identify correlates of protection. This may be a major limitation for the impact of the publication.

3. Another important limitation of the study is that mRNA primed animals are Mauritian cynomolgus macaque whereas Indian rhesus macaques were used for the subunit primed cohort. There are well known differences between these two species that significantly limits the possibility to compare the impact on the boost of the two priming strategies. This should be clearly indicated in the results section therefore warning the reader that this comparison cannot be formally performed. Following is an example of discussion that is not permitted by the study design: looking at figure 2 it appears that animals primed with D614 CoV2 preS dTM-AS03 developed in average higher responses after Beta CoV2 preS dTM-AS03 vaccine when compared to mRNA-primed animals.

4. Line 94: "At 6 months, 100% of the 94 macaques immunized with the Beta booster still had detectable nAb titers (>LOQ) against all 95 VOCs and SARS-CoV-1" ...: Figure 2 shows that this is true only for macaques that received the Beta booster only, at 5 µg. Macaques that received the combo boost with D61 and Beta-AS03 proteins seems to develop lower responses and at least one of the animals has nAb below the LOQ. A dose effect may explain this difference since lower amounts of each protein was used (2.5µg+2.5µg).

5. We understand that at the time of the manuscript submission BA.1 VOC was still of high relevance, however it would be interesting to updated cross-reactive data with new stains like BQ.1 which predominate now in most part of the world and seems to escape many of previous immunity.

6. It would be of interest as a way to increase the possibility to compare the results with other published data, to report in supplemental material the level of spike specific binding antibodies measured with standard assay.

Roger LE GRAND

Reviewer #2 (Remarks to the Author):

In this work, Pavot and colleagues report on new data from a previously published study in which NHPs were vaccinated with 2 proprietary SARS-CoV-2 vaccine technologies, based on either mRNA or spike

subunit, and boosted with subunit vaccines based on spike of the parental D614G strain, the Beta variant (B.1.351), or a mixture thereof.

The new data presented include neutralization data against different SARS-CoV-2 variants and SARS-CoV-1, 6 months post booster, as well as B cell elispot data. Boosters containing Beta spike appear to induce equivalent or higher levels of nAbs and memory B cells, although no statistical significance is reported. Overall, the study shows that the subunit vaccine technology may be a valuable addition to the arsenal of COVID countermeasures, and that immunity induced by Beta spike offers good coverage of the currently known SARS-CoV-2 variants.

Although it is my personal opinion that the novelty of these findings is debatable, given the previous report by Pavot et al. and the fact that the data presented in the current manuscript are in line with expectations, the study is scientifically sound and can be published as such.

RESPONSES TO REVIEWER COMMENTS

Reviewer #1 (Remarks to the Author):

Pavot et al reported the use of a preS stabilized recombinant spike protein of SARS-CoV-2 formulated in AS03 adjuvant (CoV2 preS dTM-AS03) as a booster vaccine against COVID-19. The work is the continuation of previous description of CoV2 preS dTM-AS03 vaccine and aims at demonstrate the broadness and durability of the induced neutralizing antibodies (nAb) in serum. For that purpose, three groups of non-human primates (NHP) primed with ancestral D614 spike expressing mRNA (two injections 4 weeks apart) were boosted six months later with either D614 CoV2 preS dTM-AS03, or Beta strain CoV2 preS dTM-AS03, or a mix of these two proteins. In addition, three other groups of NHP primed with two injections (4 weeks apart) of D614 CoV2 preS dTM-AS03, were boosted six months later as above with the same three approaches. The inclusion of Beta variant spike in the booster strategy represents a rational approach due to its immunogenicity and the potential capacity to elicit antibodies neutralizing an extended diversity of variants of concerns (VOC) which seems to be confirmed by the reported results. The authors demonstrate that 6 months post boost, most the NHP maintained significant levels of serum neutralizing activities against Alpha, Beta, Delta and BA.1 VOC. Remarkably, a majority of animals also developed nAb against more distant SARS-CoV-1. Circulating memory B cells in the blood that are specific of the spike protein have been assessed as a mechanism to explore the capacity of the vaccine to induce durable responses. Interestingly, spike specific B cells still datable six months post prime. Booster injection with CoV2 preS dTM-AS03 appeared to induce an increase of the specific IgG secreting cells in all groups at similar levels, including in animals with pre-boost low frequency of cells. This also confirms that protein boost can improve vaccine response whatever the priming strategy. The data presented here is certainly relevant to field and deserves to be considered for publication since the vaccine is already at advanced stages of the development and because the urgent need for new vaccines capable of inducing sustained cross-protective responses while avoiding multiple booster vaccine injections per year.

Specific comments:

1. In mRNA-primed macaques, at M6 post boost, GMT ranged from 19 to 1951 depending on the tested variant. Not surprisingly lowest responses are observed against BA.1 VOC. What is the threshold of neutralizing antibodies the authors consider critical to maintain a protective response? This may be at least commented in the discussion section by comparing to the large amount of currently published data using similar classical pseudotyped lentivirus based assay for testing neutralization.

Response:

The U.S. government's COVID-19 Vaccine Correlates of Protection Program assessed correlates of protection (CoPs) in phase 3 trials of four vaccines (mRNA-1273, Ad26.COV2.S, NVX-CoV2373, ChAdOx1 nCoV-19). Vaccine efficacy always markedly increased with the titer.

Meta-analyses have established high correlations between the standardized mean titer and vaccine efficacy, and the neutralizing antibody titer has consistently been shown to be a mechanistic CoP in challenge studies in nonhuman primates (McMahan et al Nature 2020; Corbett et al. Science 2021).

The evidence strongly supports the designation of the neutralizing antibody titer as a CoP.

However, according to Gilbert et al. NEJM 2022 (link), for infections in which viremia is key to pathogenesis (e.g., polio), we can identify a threshold because sufficient levels of antibody can prevent dissemination of the pathogen through the bloodstream. The same does not hold for Covid-19, since it is caused by a mucosal infection that can be invasive. Although a threshold CoP is ideal because it can provide an absolute benchmark for approving a vaccine without the need for a comparator vaccine, this goal is probably unattainable for Covid-19, because the amount of virus to which trial participants are exposed varies widely and because CoPs must be capable of predicting vaccine efficacy over a period of postvaccination follow-up during which antibody levels decline. These factors insert uncontrollable variability into the analysis such that even if neutralizing antibodies were a perfect mechanistic CoP, the titer values among people with Covid-19 would overlap with those among people without SARS-CoV-2 infection, as has been observed in all trials. Yet these partial separations, combined with evidence from all five types of sources defined above, can validate a “nonthreshold CoP” where COVID-19 risk decreased incrementally with increasing increments in antibody level.

Interestingly, in the study from McMahan et al. in rhesus macaques (link) they showed that the adoptive transfer of purified IgG from convalescent macaques protects naive recipient macaques against challenge with SARS-CoV-2 in a dose-dependent fashion.

These data demonstrate that relatively low neutralizing antibody titers (threshold: pseudovirus NAb titers of 43) using similar classical pseudotyped lentivirus-based assay are sufficient for protection against SARS-CoV-2 in rhesus macaques.

Considering this threshold, we can be confident that the monovalent Beta vaccine formulation would induce protection against all tested variants up to 6 months post-booster.

We added these points in the discussion lines 186-197.

2. We may regret that animals have not been challenged by one of the recent VOC to assess protection and to try to identify correlates of protection. This may be a major limitation for the impact of the publication.

Response:

We indeed considered assessing protection against recent VOC, however the study design was not appropriate. The primary objective of the study was to assess the long-term immunogenicity of different vaccine formulations, and the number of animals available were not sufficient to adapt the design for an efficacy study (as a reminder, the animals were re-used from 2 prior independent immunogenicity studies). In addition, at the time when this study was designed, it was unknown if Omicron infection would result in significant viral load in lungs (BALs). For both reasons, we decided not to perform a viral challenge, this also allowed to repurpose the animals for other studies in the effort to reduce the number of new macaques used.

3. Another important limitation of the study is that mRNA primed animals are Mauritian cynomolgus macaque whereas Indian rhesus macaques were used for the subunit primed cohort. There are well known differences between these two species that significantly limits the possibility to compare the impact on the boost of the two priming strategies. This should be clearly indicated in the results section therefore warning the reader that this comparison cannot be formally performed. Following is an example of discussion that is not permitted by the study design: looking at figure 2 it appears

that animals primed with D614 CoV2 preS dTM-AS03 developed in average higher responses after Beta CoV2 preS dTM-AS03 vaccine when compared to mRNA-primed animals.

Response:

We agree with the reviewer that no direct comparison can be made between the two cohorts because of the differences of species, and in male/female compositions. The following has been specified in the method/statistical section: “Due to the differences in species and gender repartition between both cohorts, no direct comparison were performed on immune responses between the two cohorts”.

However, concerning species, other internal studies using the subunit vaccine candidates performed in rhesus and cynomolgus macaques showed comparable immunogenicity after a primary immunization series in both species. Similarly, other internal studies involving males and females macaques didn't show any significant effect of the gender on the vaccine immune responses. Therefore, one can assume that, if we except the primary vaccination effect, both cohorts will be similarly responsive to a booster immunization.

4. Line 94: “At 6 months, 100% of the macaques immunized with the Beta booster still had detectable nAb titers (>LOQ) against all VOCs and SARS-CoV-1” ...: Figure 2 shows that this is true only for macaques that received the Beta booster only, at 5 µg. Macaques that received the combo boost with D614 and Beta-AS03 proteins seems to develop lower responses and at least one of the animals has nAb below the LOQ. A dose effect may explain this difference since lower amounts of each protein was used (2.5µg+2.5µg).

Response:

We agree with the reviewer that the lower dose of Beta component in the bivalent (2.5 + 2.5µg) might explain the less robust responses at 6 months against BA.1 and BA.5, as it seems that the cross-reactivity to distant variant (and SARS-CoV-1) is improved with the Beta antigen when used at 5 µg in the monovalent formulation.

We added a sentence to explain this point (lines 95-96).

5. We understand that at the time of the manuscript submission BA.1 VOC was still of high relevance, however it would be interesting to updated cross-reactive data with new stains like BQ.1 which predominate now in most part of the world and seems to escape many of previous immunity.

Response:

At the time of the experiment and manuscript submission the Omicron BA.1 was indeed still of high prevalence. Subsequently in addition, we tested neutralizing titers against the new Omicron BA.4/5. The data were added in the new Fig.3 and lines 115-120. Unfortunately, we did not test cross-neutralizing titers against newer strains that continue to emerge.

6. It would be of interest as a way to increase the possibility to compare the results with other published data, to report in supplemental material the level of spike specific binding antibodies measured with standard assay.

Response:

ELISA titers were added in supplementary figure 1.

Reviewer #2 (Remarks to the Author):

In this work, Pavot and colleagues report on new data from a previously published study in which NHPs were vaccinated with 2 proprietary SARS-CoV-2 vaccine technologies, based on either mRNA or spike subunit, and boosted with subunit vaccines based on spike of the parental D614G strain, the Beta variant (B.1.351), or a mixture thereof.

The new data presented include neutralization data against different SARS-CoV-2 variants and SARS-CoV-1, 6 months post booster, as well as B cell elispot data. Boosters containing Beta spike appear to induce equivalent or higher levels of nAbs and memory B cells, although no statistical significance is reported. Overall, the study shows that the subunit vaccine technology may be a valuable addition to the arsenal of COVID countermeasures, and that immunity induced by Beta spike offers good coverage of the currently known SARS-CoV-2 variants.

Although it is my personal opinion that the novelty of these findings is debatable, given the previous report by Pavot et al. and the fact that the data presented in the current manuscript are in line with expectations, the study is scientifically sound and can be published as such.

Response:

We thank the reviewer for these comments.

Regarding the novelty, we think those data are very important for the scientific and medical community since they show the full neutralizing antibody kinetics all along the 6 months follow up after booster and against 4 SARS-CoV2 variants of concern and SARS-CoV1. These data were not available at the time of the previous report and add strong input to the current paper showing the longevity of the circulating antibodies as well as Spike-specific memory B cell responses.

REVIEWERS' COMMENTS

Reviewer #1 (Remarks to the Author):

I have very much appreciated the quality of the responses and the complement of information added to the manuscript.

I have no further requests,

Roger Le Grand